# Comprehensive Analysis of GABA_A_-A1R Developmental Alterations in Rett Syndrome: Setting the Focus for Therapeutic Targets in the Time Frame of the Disease

**DOI:** 10.3390/ijms21020518

**Published:** 2020-01-14

**Authors:** Alfonso Oyarzabal, Clara Xiol, Alba Aina Castells, Cristina Grau, Mar O’Callaghan, Guerau Fernández, Soledad Alcántara, Mercè Pineda, Judith Armstrong, Xavier Altafaj, Angels García-Cazorla

**Affiliations:** 1Synaptic Metabolism Lab, Neurology Department, Institut Pediàtric de Recerca, Hospital Sant Joan de Déu and CIBERER, 08950 Barcelona, Spain; cgrau@fsjd.org (C.G.); agarcia@sjdhospitalbarcelona.org (A.G.-C.); 2Genetics Department, Institut Pediàtric de Recerca Hospital Sant Joan de Déu, 08950 Barcelona, Spain; cxiol@sjdhospitalbarcelona.org (C.X.); acastells@fsjd.org (A.A.C.); gfernandezi@sjdhospitalbarcelona.org (G.F.); jarmstrong@sjdhospitalbarcelona.org (J.A.); 3Neurology Department, Institut Pediàtric de Recerca Hospital Sant Joan de Déu, and CIBERER, 08950 Barcelona, Spain; mocallaghan@sjdhospitalbarcelona.org (M.O.); pineda@sjdhospitalbarcelona.org (M.P.); 4Neural Development Lab, Departament de Patologia i Terapèutica Experimental, Institut de Neurociènces, Universitat de Barcelona, IDIBELL, 08950 Barcelona, Spain; salcantara@ub.edu; 5Bellvitge Biomedical Research Institute, Neuropharmacology and Pain Unit, University of Barcelona, 08950 Barcelona, Spain

**Keywords:** Rett syndrome, GABA, autism, GABA-A1R, KCC2, RNAseq

## Abstract

Rett syndrome, a serious neurodevelopmental disorder, has been associated with an altered expression of different synaptic-related proteins and aberrant glutamatergic and γ-aminobutyric acid (GABA)ergic neurotransmission. Despite its severity, it lacks a therapeutic option. Through this work we aimed to define the relationship between MeCP2 and GABAA.-A1 receptor expression, emphasizing the time dependence of such relationship. For this, we analyzed the expression of the ionotropic receptor subunit in different MeCP2 gene-dosage and developmental conditions, in cells lines, and in primary cultured neurons, as well as in different developmental stages of a Rett mouse model. Further, RNAseq and systems biology analysis was performed from post-mortem brain biopsies of Rett patients. We observed that the modulation of the MeCP2 expression in cellular models (both Neuro2a (N2A) cells and primary neuronal cultures) revealed a MeCP2 positive effect on the GABAA.-A1 receptor subunit expression, which did not occur in other proteins such as KCC2 (Potassium-chloride channel, member 5). In the Mecp2+/− mouse brain, both the KCC2 and GABA subunits expression were developmentally regulated, with a decreased expression during the pre-symptomatic stage, while the expression was variable in the adult symptomatic mice. Finally, the expression of the gamma-aminobutyric acid (GABA) receptor-related synaptic proteins from the postmortem brain biopsies of two Rett patients was evaluated, specifically revealing the GABA A1R subunit overexpression. The identification of the molecular changes along with the Rett syndrome prodromic stages strongly endorses the importance of time frame when addressing this disease, supporting the need for a neurotransmission-targeted early therapeutic intervention.

## 1. Introduction

Rett syndrome (RTT; OMIM #312750) is a severe neurodevelopmental disorder characterized by a regression in the neurological development between 6 and 18 months following a normal early development. Patients experience seizures, autistic features, apnea/hyperpnoea and a loss of all the acquired capabilities, speech and non-verbal communication capacity, stereotypes, loss of purposeful use of hands, and organic dysfunctions [1,2].

The complexity of Rett syndrome derives from the MeCP2 protein function, coded by *MECP2* (Xq28; MIM* 300005), as most reported cases are associated with its defective activity. MeCP2 is a nuclear protein that acts as an epigenetic regulator, controlling the expression of numerous genes (either as transcription activators or repressors) involved in several biological processes [3]. Whilst it is a ubiquitous protein, MeCP2 is most highly expressed in the brain [2,4], most precisely in post mitotic neurons [5,6], and its deficiency results in a global neurodevelopment disturbance [7]. Neurochemically, Rett syndrome has been associated with an aberrant expression of neurotransmitters, neuromodulators, transporters, and receptors [8,9,10,11]. Collectively, these alterations may underlie an unbalanced excitatory/inhibitory neurotransmission together with a disturbed synaptic development associated with Rett syndrome [12,13]. In particular, an unbalanced excitatory/inhibitory neurotransmission stands out, with a specific γ-aminobutyric acid (GABA)ergic malfunction. GABA (γ-aminobutyric acid) is the major inhibitory neurotransmitter in the brain [14]. The fast inhibitory actions of GABA are mediated by the GABA(_A_) receptors, which are ligand-gated chloride (Cl-) channels consisting on assemblies of five different subunits from eight possible subfamilies [15], the 2α1 + 2β2 + 1γ2 conformation being the most prevalent, accounting for 43% of the total GABA_A_ receptors [16], present in most brain areas. The selective transport of Cl^−^ when the GABA_A_ receptors are activated hyperpolarizes the neuron, reducing its likelihood of starting an action potential [17].

GABAergic synapses dysfunction has been associated with several Rett features. This altered performance, nevertheless, seems to be region and developmental-stage dependent. In fact, studies in MeCP2-/y mice brain slices show reduced miniature excitatory postsynaptic currents in the somatosensory cortical neurons, together with unaltered miniature inhibitory postsynaptic currents, which result in an overall reduced excitation. Opposite to that, there is a reduced conductance but increased excitatory/inhibitory ratio in the CA1 and CA3 areas of the hippocampus and V1 pyramidal neurons in in vivo visually evoked responses. Many players appear to be participating in this GABAergic neurotransmission alteration, from GABA receptors [18,19,20] to the chloride channels NKCC1 and KCC2 [13,21], responsible for the excitatory to inhibitory switch of GABAergic synapses during development, and for which the expression has been found to be altered in Rett patients’ cerebrospinal fluid (CSF).

In agreement with this, the restoration of correct GABAergic neurotransmission partially rescued Rett-like phenotypic abnormalities in mouse models [22], supporting the GABAergic pathway pivotal role in Rett’s pathophysiology, and opening a window for the treatment-expectancy of the disease [23].

Regarding the need for a treatment for the disease, and proving its mentioned potential reversibility, neurotransmission modulation appears to be an attractive therapeutic approach. However, enlightened by very last reports, the question of “when” adds on to the “how” for treating the disease. Given the severity of Rett syndrome and the lack of therapeutic options, there is an urge for the definition of the molecular alterations during development that set the bases for the window travel to address novel therapeutic targets, as intended throughout this work.

Throughout this work, we aimed to define if there was a direct relationship between GABAergic synapses known to alter elements, and MeCP2. Our results show a direct relationship between MeCP2 and GABA ionotropic receptors’ expression, which is not extensible to other GABAergic proteins such as KCC2, altered in the context of MeCP2 dysfunction, but not appearing to be directly regulated by its activity. More important than this, our work points attention towards the importance of the time frame when addressing Rett syndrome, as changes appear to be time-dependent, with greater importance for the pre-symptomatic stages. 

## 2. Results

### 2.1. MeCP2 Levels Are Associated with GABA_A_ Receptors’ Expression in Cellular Models

Fast GABAergic neurotransmission is mediated by GABA ionotropic receptors (GABA_A_). These are ligand-gated chloride (Cl^−^) channels consisting on five subunits of eight subfamilies [15]. Mechanistically, GABA_A_ receptors’ activation allows for a selective Cl^−^ influx, triggering a hyperpolarization of the postsynaptic neuron that reduces its likelihood of starting an action potential [17]. Among the multiple stoichiometric combinations of the heteromeric GABA_A_ receptors, α1-β2-γ2 is the major molecular combination, with the α1 subunit being present in over 60% of GABA_A_ heteromers, being widely expressed in brain areas.

Based on previous reports supporting the contribution of GABAergic dysfunction in Rett syndrome progression and the alteration of the GABA_A_-A1 expression, we hypothesized that MeCP2 disturbance might directly affect the density of GABA_A_ receptors, rather than such an altered expression being a secondary effect of an overall GABAergic dysfunction. To this end, we studied its expression pattern over two days of Mecp2 knockdown in cellular models.

On one side, we interfered mMeCP2 (mouse *MECP2*) expression in Neuro2a (N2A) mouse neuroblastoma cells by the transient transfection of a shRNA-anti-3′UTR Mecp2 plasmid (from now on, mentioned solely as shRNA). Prior to the GABA_A_-A1R assessment, the efficiency and specificity of the *MECP2* expression interference was evaluated. For this, the total RNA was extracted from the cells under the four scenarios analyzed, that is, un-transfected N2A cells, cells transfected with the shRNA vector or co-transfected with the shRNA vector, and the *hMECP2* gene in either its wild type (wt) or mutated versions. Interestingly, the short hairpin sequence was designed to specifically target mouse Mecp2, while the human MECP2 (hMeCP2) was predicted to be insensitive to shRNA-anti-(3′UTR)Mecp2. The system was proven to be reliable, as the endogenous *MECP2* levels were drastically reduced upon the transfection of N2A cells with the shRNA vector, while the exogenous expression of the human isoform, however, was not affected by the shRNA co-expression. As shown in Figure 1A, the amplification with specific *hMECP2* primers revealed an increased expression only upon transfection with the *hMECP2* wt vector.

While the specificity of the shRNA interference has been shown, the exclusivity of the hMeCP2 cannot be ensured, as endogenous mMeCP2 is amplified in the un-transfected N2A cells using hMeCP2 primers (Figure 1A, right panel). We believe we are detecting undesired amplification due to cross-binding between the mouse and human forms. Human and mouse MeCP2 are identical in 87.41% of their RNA sequence. While the forward primer binds to a sequence similar in 95% of both forms (19 out of 20 bases), the reverse primer was designed to only bind to the human transcript, which, however, seems not to have been properly achieved.

Under basal conditions, the transcript encoding for the major GABA_A_ receptors subunit (*Gabra1*) was not detectable in N2A cells. Remarkably, the target was amplified upon transfection with the plasmid carrying the *wt* form of *hMECP2* (Figure 1B), supporting a relationship between MeCP2 expression and the gene over-expression. Noteworthy, the GABA_A_ receptor subunit expression was undetectable upon transfection with the plasmid containing *hMECP2*_c.763C>T (Figure 1B). The same happened with the remaining principal GABA_A_ subunits *Gabrb2* and *Gabrg2* (data not shown). These results point towards a direct regulation of GABA_A_-A1R expression by MeCP2 during, at least, a certain time frame of development.

Interestingly, with the KCC2 expression, which has been reported to be affected in Rett syndrome models, patients did not appear to be affected by MeCP2 inhibition, as can be shown in Figure 1B, suggesting that both alterations affecting the same synapse occur through different mechanisms.

In order to evaluate the identified MeCP2 positive regulatory effect on the GABA_A_ receptor expression in the dendritic processes, the primary cultures of murine cortical neurons were established. Following shRNA-mediated MeCP2 silencing, the expression of the *GABRA1* subunit was assessed.

The primary cortical neurons were transiently co-transfected at day *in vitro* 7 (DIV7) with either a shRNA-anti-(3′UTR)Mecp2 or a mock plasmid, together with a pcDNA-EGFP vector in a 1:7 ratio, allowing for the identification of shRNA-transfected neurons. An immunofluorescence analysis was performed in the early mature primary neuronal cultures (DIV11), showing a complete lack of Mecp2 detection in shRNA-(GFP-positive) transfected neurons (Figure 2A), and thus validating the inhibition system.

When we next evaluated the GABA_A_-A1R expression on the silenced neurons, we observed a severe decrease in the detection in the GFP-positive neurons, compared with the cultures transfected with GFP and mock vector (Figure 2B), considered as the control conditions. The mean detection on the GABRA1 marking was 37% compared with the control conditions. These results are complementary to the N2A cells observations, as both support a positive and straightforward relationship between MeCP2 and GABA_A_-A1 receptor expression.

### 2.2. Neurodevelopmental Changes of GABA_A_-A1R and KCC2 in a Rett Syndrome Mouse Model Point Towards the Importance of Pre-Symptomatic Versus Symptomatic Manifestations

As Rett syndrome is a neurodevelopmental disorder, we aimed to appraise the studied changes in different evolutionary stages.

Thus, we evaluated the GABA_A_ A1 subunit expression and KCC2 in the cortex of mice at pre-sympromatic and symptomatic stages (one and six months, respectively). As aforementioned, we selected these two proteins for their proven implication in Rett syndrome pathophysiology. We used female Rett mice models from the Bird strain, which recapitulate Rett-like abnormalities [24]. Western blot analysis revealed a significant decrease of the ionotropic GABA receptor subunit at a young stage (one month-old), while no significant differences were detected between the genotypes (Figure 2) at the symptomatic stage (six months old).

To contextualize this difference between the prodromic stages and considering whether it was a GABA_A_ A1 receptor subunit specific variation or a generalized GABAergic synapse downregulation, we analyzed two other proteins expression in the same conditions, namely: KCC2 and GAD67. The first one, KCC2, is a neuron-specific chloride potassium symporter responsible for the maintenance of intracellular chloride concentrations. On the other side, GAD67 or glutamate decarboxylase is the enzyme that catalyzes the decarboxylation of glutamate to GABA and CO^2^, and is widely used as a GABAergic marker. As can be seen in Figure 2B, a decreased expression of both markers was observed in the pre-symptomatic stage, suggesting a generalized decrease of the GABAergic function.

Parallel to this, and as a control, the MeCP2 expression was assessed in the same samples and appeared to be lower in all of the Rett samples (Figure 3).

### 2.3. Transcriptomic Profile of the GABAergic Pathway in Post-Mortem Brain of Rett Syndrome Patients Shows a Generalized GABAergic Pathway Upregulation

The above-mentioned results allowed for the definition of the MeCP2-related expression of GABA_A_-A1 R, highlighting the difference between pre-symptomatic and phenotypic stages. Furthermore, we evaluated the expression of the 108 genes shaping the GABAergic synaptic pathway, as defined in the Kegg pathways (Kegg pathway: map04727).

Upon the validation of the RNAseq experiment, as described in Materials and Methods, and because of the lack of healthy control samples, the raw data from the Rett patients’ necropsies were compared with the RNAseq data of the control individuals, available at GTEx Portal [25]. We crossed our data with the data derived from five controls (two females and three males), all aged 20 to 29, being the closest age-matched group to our patients (10 to 15 years old at exitus). All of the data were normalized with housekeeping genes *RPLP0* and *GUSB*.

The transcripts encoding for the major GABA_A_ receptor subunits, GABRA1, and the chloride transporter KCC2, were slightly increased in the patients’ samples compared with the controls (Figure 4). Similar as expected from the Western blot studies, the GAD67 levels remained unaltered compared with the controls. 

## 3. Discussion

The present work arises from the need to deepen into the definition of Rett’s syndrome pathophysiology in order to define new therapeutic strategies. To such end, we have explored the GABAergic neurotransmission system in different evolutionary stages of the disease, setting the focus on the main GABA ionotropic receptor, GABA_A_-A1R. We have observed a direct relationship between the MeCP2 altered expression and GABAergic receptors disruption, which is strongly dependent on the prodromic stage of the disease, angling the focus towards the time frame, which will be a key factor when looking for therapeutic options.

The triangle conformed by Rett syndrome, MeCP2, and GABAergic synapses has been previously explored by other groups [9,12,26]. An increased MeCP2 expression in GABAergic neurons has been reported, and a reduced GABA release was reported upon MeCP2 knocking out in forebrain GABAergic neurons [27]. Such a relationship has been shown to be extensive to other neuron types and brain areas, such as CA3 hippocampal neurons or brainstems [12]. Moreover, in 2016, Dr. Zoghbi’s team showed how the restoration of the MeCP2 expression exclusively in GABAergic neurons was sufficient to rescue some disease features in a mouse model of Rett syndrome [28]. Even a time-dependent alteration was suggested in male mice by the authors of [26]. Through our work, we have followed such a line of thought, wondering how the alterations were occurring during development, setting the focus on the pre-symptomatic stages.

We first wondered if there was a direct relationship between MeCP2 and the postsynaptic GABA_A_-A1R expression. Previous studies pointing out a relationship between GABA receptors and MeCP2 had been done in the context of whole brain models’ analysis, so it remained unclear if the potential decrease in GABA receptors was strictly related to MeCP2 expression or was a secondary effect to a global dysfunction. To address that question, we performed two complementary experiments. On first term, we overexpressed MeCP2 in a cell system that almost did not express GABA ionotropic receptors (or its expression was mostly below our detection sensitive), namely N2A cells. To increase the assay restricted conditions, we silenced any potential endogenous MeCP2 expression, and overexpressed the human form in either its wild-type or mutated forms. As expected, the GABA ionotropic receptors were only detected upon transfection with the MeCP2 *wt* form, not occurring when transfected with the mutated version. These results were complemented with the peer experiment in primary cortical neurons; this time, we imaged a decrease in GABA_A_-A1R expression four days after silencing the MeCP2 expression. Our results strongly suggest a direct relationship between MeCP2 and post-synaptic GABA ionotropic receptors’ expression, rather than this being a secondary effect of an overall altered homeostasis. Opposite to that, we found that the KCC2 expression was unaltered in the presented scenario. KCC2 is a chloride channel, essential for GABAergic correct functioning, and described to be down regulated in Rett syndrome [13]. The fact that its expression was not significantly altered by MeCP2 inhibition suggests that such an altered regulation is more related to the syndrome pathophysiology rather than to the straightforward MeCP2 mutations. In fact, it has been well described how the KCC2 expression and function is, indeed, regulated by the GABA function itself [29,30].

It is largely known that most GABA_A_ receptor coding genes are clustered in four chromosomal regions in chromosomes 4, 5, 15, and 19 [31]. These subunits comprising the pentameric GABA receptor formation have a coordinated expression [32], and, as revealed by human brain transcriptome analysis, this produces a subject and region-specific expression signature of GABA_A_ receptor subunits [33]. Enlightened by our results, further studies should be made to elucidate whether MeCP2 acts a transcriptional regulator of these clusters, the mechanisms through which this regulation takes place, and the time during development.

As stated from the beginning of the discussion, our main objective was setting the focus over the evolutionary stages in the disease. As for that, we switched to a model that allowed us to evaluate the different prodromic stages of the disease. We used females Bird’s Rett mice model, as they better recapitulate the pathophysiology [34]. We focused our analysis on the following three proteins: GABA_A_-A1R, KCC2, and GAD67. The first two proteins were selected because of the previous observations, and because they have been proven to be not only crucial in Rett syndrome development, but also potential actionable targets of the disease [21,35]. As shown by our results, a markedly reduced expression of GABA_A_-A1R and KCC2 was recorded in pre-symptomatic mice, while the GAD67 population (used as a marker for GABAergic neuronal) remained unaltered. These results suggest a reduced GABAergic activity without the affectation of the GABAergic general population, which are aligned with the previously described results. Expanding previous descriptions, our results bring focus to the pre-symptomatic stages of the disease, where the most differences were observed. Furthermore, we did not observe any reliable difference in both proteins’ expressions in fully-symptomatic mice (or even an increase in KCC2 expression), once again, enhancing the importance of the time-frame when addressing Rett syndrome. The variability in KCC2 expression, and its activation through phosphorylation, is a field that further studies should explore, especially under the recent scenario in which KCC2 is being addressed as a therapeutic target [36]. Preliminary results (data not shown) have pointed towards an over-phosphorylation of KCC2 in symptomatic mice, drawing a scenario in which KCC2 will be under expressed in early pre-symptomatic stages and inactivated in symptomatic phases—again, shaping different therapeutic strategies on different prodromic stages. KCC2 activation and membrane diffusion has been related with GABAergic activity itself, increasing its therapeutic interest [29,30].

During the preparation of this manuscript, Dr. Ben-Ari’s team also pointed out the importance of GABAergic dysfunction during development in Rett syndrome [37], providing neurophysiological evidence of such a temporal alteration. The detected expression to normal extents of GABA_A_-A1R in mature Rett brains reinforces the idea of time-dependence in the MeCP2 control of the GABAergic cluster during the specific developmental stages.

Completing the observed results, we analyzed the aforementioned targets in two Rett patient’s necropsied brains. Backing up our previous description, the results showed an even enhanced expression of GABA_A_-A1R and KCC2, without alterations on GAD67. The overall analysis of the GABAergic pathway showed a slightly increased expression of almost all of the implicated genes. These results confirm the previous findings, pointing towards the importance of pre-symptomatic damage. These results are in agreement with the dataset reported by Renthal et al. [38]. An increasing body of evidence pointing towards the importance of early intervention has been reported in the last few years, as reviewed by Constentino et al. [39], and has extended from neurotransmission to other therapeutic targets in Rett syndrome, such as energetic dysfunction, as very recently published [40], or inflammatory processes [41,42].

To summarize, our results show a direct relationship between MeCP2 and GABA ionotropic receptors’ expression, which is not extensible to other GABAergic proteins such as KCC2, which is altered in the context of MeCP2 dysfunction, but does not appear to be directly regulated by its activity. More important than this, our work points attention towards the importance of the time frame when addressing Rett syndrome, as changes appear to be time-dependent, with greater importance in the pre-symptomatic stages.

Therapeutically, early GABAergic modulation in Rett syndrome may represent a promising strategy. While our results suggest that GABA-A1 R can be a potential therapeutic target, the time window of intervention is, according to our findings, critical. Additionally, the development of novel drugs enhancing GABA-A1 R function (for potential use in the initial clinical stages) and devoid of side effects are required, for an early intervention of Rett syndrome.

## 4. Materials and Methods

### 4.1. Cell Lines and Samples Utilization

Immortalized Neuro2a cells (also known N2A cells, a fast-growing mouse neuroblastoma cell line) were grown following standard conditions in Dulbecco’s Modified Eagle’s Medium (DMEM) supplemented with 1% glutamine, 10% fetal bovine serum (FBS), and antibiotics.

For neuronal primary cells cultures, the protocol described in the literature [43] was followed. All of the experimental procedures were carried out according to European Union guidelines (Directive 2010/63/EU) and following protocols that were approved by the Ethics Committee of the Bellvitge Biomedical Research Institute (IDIBELL, Barcelona, Spain). Briefly, mouse embryos (embryonic day E18) were obtained from pregnant CD1 females, the cortexes were isolated and maintained in cold Hank’s Balanced Salt Solution supplemented with 0.45% glucose (HBSS-Glucose) and digested mildly with trypsin for 17 min at 37 °C, and dissociated. The cells were washed three times in HBSS and resuspended in a Neurobasal medium supplemented with 2 mM Glutamax (Gibco, Waltham, MA, USA) before filtering in 70 mm mesh filters (BD Falcon, San Jose, CA, USA). The cells were then plated onto glass coverslips (5 × 10^4^ cells/cm^2^) coated with 0.1 mg/mL poly-L-lysine (Sigma, Darmstadt, Germany), and 2 h after seeding, the plating medium was substituted by a complete growth medium, consisting of a Neurobasal medium supplemented with 2% B27 (Invitrogen, Waltham, MA, USA) and 2 mM Glutamax.

In this study, we used post mortem brain samples from two unrelated Rett patients bearing the *MECP2*_c.763C>T mutation and an intra-assay control for RNAseq (i.e., an extra sample that due to its pathology could not be used as a bona fide control, but that allowed us to run technical comprobations). This mutation is the second most frequent Rett-causative mutation, present in 10.9% of the cases [44]. The patients were between 10 and 15 years old at exitus, which is noteworthy, as life expectancy is not highly reduced in Rett syndrome. In all of the cases, RNA was isolated from two brain regions (frontal and occipital cortex) and the samples were treated according to the informed consent of the legal representatives.

The study was approved by the Ethics Committee of Sant Joan de Déu Hospital, project PI15/01159, 01/2016. We are indebted to the “Biobank de Hospital infantil Sant Joan de Déu per la Investigació” integrated in the “Spanish Biobank Network of ISCIII for the sample and the data procurement”.

### 4.2. Mouse Colony

Cortex samples from one and six-month old Mecp2^−/+^ female mice [45] Bird’ model (B6.129P2(C)-*Mecp2*^tm1.1Bird^/J) were obtained after mouse sacrifice and brain dissection. The proteins from cortex were extracted by tissue homogenization with an ice-cold RIPA buffer with protease inhibitors (cOmplete, mini, EDTA-free protease inhibitor cocktail, Merck), for 30 min at 4 °C followed by 15 min of centrifuge at 4 °C. The protein samples were quantified by Bradford assay and stored at −80 °C.

### 4.3. Plasmids and Mutagenesis

In certain experiments, we attempted to silence the endogenous MECP2 expression and re-express the human gene either in the wild type or mutated form.

For the MECP2 silencing, transient transfection with a vector containing a shRNA targeting mMECP2 was performed. Silencing was done with the MISSION^®^ shRNA technology (Sigma Aldrich, Darmstadt, Germany; Clone TRCN0000304464), and the efficiency was checked at the protein level. To ensure the sole silencing of the endogenous gene, and not the re-expressed form, we used a shRNA directed to the 3′UTR part of the gene, absent in the transfected cDNA. We used, as a control for transfection, a vector with the same backbone but no shRNA.

The MeCP2 c.763C>T genetic variant was introduced by site-directed mutagenesis using the QuickChange II XL Kit (Agilent Technologies, Santa Clara, CA, USA), in the pEGFP-C1-hMeCP2 (wild-type) mammalian expression vector (kindly provided by Dr. Landsberger). The mutation was confirmed by Sanger sequencing. Both vectors, together with the pEGFP-C1 vector (BD Clontech, Palo Alto, CA, USA) and a mock vector with the same backbone, were used for the experiments that required plasmid transfections.

These were carried out using Lipofectamine 2000 (Thermofisher, Waltham, MA, USA) following manufacturer recommendations. For neuronal primary cultures, 0.8 g of total DNA was mixed with Lipofectamine 2000 and incubated with cortical neurons (at DIV11). The transient expression was allowed for 96 h, and the neurons were fixed at DIV14. For the N2A cells, 4 g of DNA was transfected over 300,000 cells grown in 10 cm^2^ plates, and the cells were collected four days after transfection for RNA analysis.

### 4.4. RNA Extraction and qRT-PCR

RNA for RNAseq (from post-mortem human brain samples) and for qRT-PCR (from post-mortem human brain samples and N2A cells) was extracted using RNeasy^®^ Fibrous Tissue Mini Kit (Qiagen, Hilden, Germany), following the manufacturer’s instructions. The total RNA was eluted in 40 μL of RNAse-free water and stored at −80 °C. The RNA concentration was measured using the NanoDrop 2000 Spectrophotometer (ThermoScientific, Waltham, MA, USA), and integrity was assessed by running the samples through a 1% agarose gel.

qPCRs were carried out following a two-step protocol. First, cDNA was synthesized from a total of 500 ng of RNA per reaction, following the recommendations provided with SuperScript™ III First-Strand Synthesis SuperMix for qRT-PCR (Invitrogen^TM^). After the RT-PCR reaction, the post-mortem brain cDNA from frontal and occipital cortex samples was pooled.

Second, qPCR was performed in a QuantStudioTM 6 Flex Real Time PCR System (Applied Biosystems^TM^ MA, USA) with PowerUpTM SYBRTM Green Master Mix (Applied Biosystems^TM^). The data were analyzed using a comparative method, correlating the initial template concentration with the cycle threshold (Ct) so as to obtain the relative quantity (RQ) of the RNA. The RQ is defined as 2^-ΔΔCt^, where ΔΔCt is the ΔCt of the patient cell line minus the ΔCt of the control cell line, and ΔCt is the Ct of the target gene minus Ct of the endogenous gene (*RPLP0* and *GUSB*).

The probes design was done through the selection of the exonic areas present in all of the transcript variants of each gene, by the selection of common fragments in the UCSC genome browser, based on GRCh38/hg38 version. Primers for N2A-derived qPCR experiments were, in 5′-3′ sense, as follows: m-Mecp2 (F: ACCATCATCACCACCATCAC; R: GGGCATCTTCTCTTCTTTGC), h-MECP3 (F: AGGAGAGACTGGAAGAAAAGT; R: CTTGAGGGGTTTGTCCTTGA), m-Gabra1 (F: ACCAGTTTCGGACCAGTTTC; R: TACAGCAGAGTGCCATCCTC), m-Gabrb2 (F: TCGCTGGTTAAAGAGACGGT; R: TCTCTCCAGGCTTGCTGAAA) and m-Gabrg2 (F: TGGGCTACTTCACCATCCAG; R: GCCATACTCCACCAAAGCAG). The primers sequence for the brain samples of qPCRs were not included.

### 4.5. Western Blotting and ICC

Western blot analysis of the cortex protein samples from Mecp2^−/+^ female mice was performed. The proteins were subjected to SDS-PAGE and transferred to a nitrocellulose membrane using the Pierce^®^ Power Station (Thermo Scientific). The membranes were blocked with milk, as follows: PBST buffer 5% for 1 h at room temperature. Incubation with primary antibodies was directed against GABA-A1 (Neuromab, UCDavis, CA, USA, 75-136) at a concentration of 1:500, MeCP2 (ab2828; Abcam, Madrid, Spain) at a concentration of 1:1000, and vinculin (ab129002, Abcam, Madrid, Spain) was performed o/n. The secondary antibodies used were horseradish peroxidase-conjugated goat anti-rabbit and goat anti-mouse IgG antibodies (Lfe Technologies, Waltham, MA, USA); these were detected using the Enhanced Chemiluminescence System (GE Healthcare, Berkshire, UK).

Immunochemistry experiments were performed as described in the literature [43]. Anti-GABA A1R (Neuromab, UCDavis, CA, USA, 75-136) was used at a concentration of 1:100, and MeCP2 (ab2828; Abcam, Spain) at a concentration of 1:250. The conjugated secondary antibodies for the confocal microscopy were used.

Fluorescence was visualized with a Leica TCS-SL spectral confocal microscope (Leica Microsystems, Wetzlar, Germany) using a Plan-Apochromat 63×/1.4 N.A. immersion oil objective (Leica Microsystems). To excite the different fluorophores, the confocal system was equipped with excitation laser beams at 488 and 546 nm. The images were analyzed with ImageJ software. For the intensity quantification of the ICCs, pictures from independent primary cultures were used. Regions Of Interest (ROIs) were defined on the green channel (GFP positive neurons) applying a threshold to only select the desired neuron. The same ROI was exported to the red channel pictures, and the mean gray value was measured.

### 4.6. RNAseq Data Analysis

Origin of data: A frontal and occipital cortex paired-end RNAseq was performed for two RTT patients bearing the *MECP2*c.763C>T, p.Arg255* mutation and the mentioned intra-assay control. Technical triplicates (three separate RNA extractions) were run for each of the two brain regions. The RNA samples were sent to the Centre Nacional d’Anàlisi Genòmica (CNAG), where the RNAseq experiment was performed, within the framework of the project FIS PI15/01159 Rett Syndrome (IP: Judith Armstrong, Ph.D.). Both brain areas were sequenced separately, and as no significant differences were observed between them, they were analyzed as a sole data pool. To discard the differences between the brain areas, we performed a principal component analysis (PCA). In such an analysis, we compare the variance between all of the samples (patients and areas). At this point, we also added an “intra-assay control” sample. Principal Component 1 (explaining 75% of the variance), clearly discriminated between the Rett and not-Rett samples, while there was not a principal component separating the brain areas (data not shown).

Because of the lack of true control data, the data from our RNAseq were compared to various public controls’ data. We used data from public controls available on the GTEx (The Genotype-Tissue Expression) Portal. Cortex RNAseq data from five controls were used, two of which were female (GTEX-15ER7 and GTEX-T2IS) and three were male (GTEX-12126, GTEX-T5JC, and GTEX-WHSE), all of them with ages ranging 20–29. We also compared our data to the cortex RNAseq data from two public male controls (ages 24 and 39) available on the Allen Human Brain Atlas, and the RNAseq data from three female control samples (ages 15–25) used in a publication by Lin, et al. in 2016 (3). Two of these samples were a pool of frontal and temporal cortex RNA, and one of them was just temporal cortex RNA.

RNAseq analysis pipeline: As a result of the low performance of the sequencing experiment, the internal control’s occipital cortex data was excluded from the analysis. The RNAseq analysis pipeline was run by the Bioinformatics Unit from the Genetics and Molecular Medicine Service at the Hospital SJD. The FASTQ files passed through a first quality control, after which a trimming was performed and the adapters were removed. Then, low quality bases were eliminated so only reads longer than 70 bp were left to continue the analysis. Here, a second quality control was performed and the reads were mapped with TopHat2 (4). The counting was performed with HTseq (5) and the R package DESeq2 (6) was used for library normalization. The frontal and occipital cortex data from our two RTT patients were averaged.

In order to compare the data obtained from our RNAseq experiment to the public data, an internal normalization over the endogens RPLP0 or GUSB of every patient’s and control’s data was performed.

A validation of the results was carried out at the Hospital Sant Joan de Déu using qRT-PCR, comparing patients with an internal RNAseq control; that is, a sample that could be used for a later comparison of the results through qPCR and therefore RNAseq technical validation, but could not be used to biologically validate the results, as it was not a healthy brain, as previously described. Thus, following the RNAseq analysis, the targeted gene expression of a subset of 21 genes corresponding to different nodes of the GABAergic pathway and differentially expressed between patients and the internal control was validated by qRT-PCR, showing a strong correlation (20 out of 21 transcripts deregulated; Appendix A), with an overall coincidence between RQ values (qRT-PCR experiments) and fold-change (RNAseq experiment).

### 4.7. Data Availability Statement

The present study is not a clinical trial of any kind. All of the data, materials, and methods to conduct the research are available in the manuscript. Patient samples are located at the “Biobank de Hospital Infantil Sant Joan de Déu per la Investigació” integrated in the “Spanish Biobank Network of ISCIII for the sample and the data procurement”, to whom we are indebted.

## Figures and Tables

**Figure 1 ijms-21-00518-f001:**
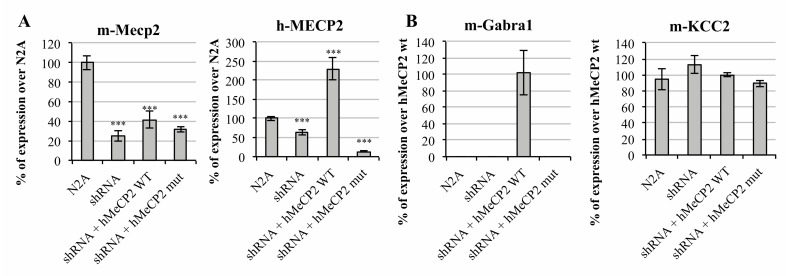
In vitro analysis of MeCP2 altered activity over γ-aminobutyric acid (GABA) ionotropic receptors’ expression. Bar graph representing the relative expression of (**A**) mouse MECP2 (mMeCP2), human MECP2 (hMeCP2) and (**B**) mouse GABRA1, and mouse KCC2, measured by qRT-PCR under four different transfection scenarios, namely: non-transfected cells (N2A), transfected with the shRNA-anti-(3′UTR)Mecp2 (shRNA) and co-transfected with the shRNA-anti-(3′UTR)Mecp2 and *wt*, or c.763C>T mutated MeCP2 carrying plasmids. Error bars represent the standard deviation of the average values. Statistical significance was calculated through a Student’s t-test (*** *p*-value < 0.001).

**Figure 2 ijms-21-00518-f002:**
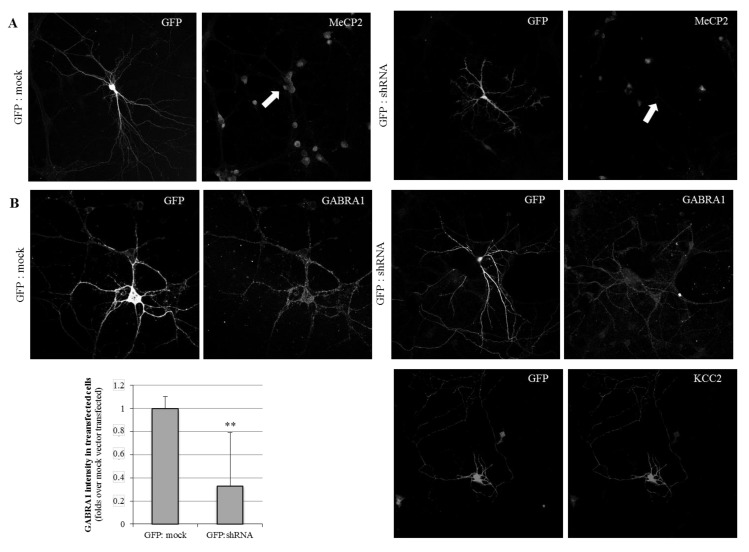
Immunofluorescence analysis of MeCP2, GABRA1, and KCC2 expression in cortical primary neuronal cultures. Cells were transfected with either GFP and mock DNA, or GFP and shRNA-anti-(3′UTR)Mecp2. Images show neurons at DIV11. Images were taken at 63x with constant time of exposure. Transfected neurons were labeled with anti-GFP and anti-Mecp2 (**A**). For the GABRA1 and KCC2 immunostaining (**B**), different neurons are shown. The quantification of the mean GABRA1 immunosignal in mock and shRNA transfection conditions is shown in the bar graph. n = 10 different neurons, from two independent cultures. ** refer to *p*-value < 0.01.

**Figure 3 ijms-21-00518-f003:**
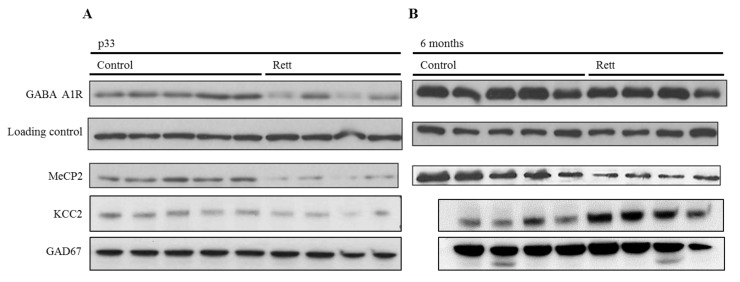
Developmental expression analysis of GABAergic proteins in the MeCP2^−/+^ mouse brain cortex. Representative Western blot analysis of the GABA_A_-A1R, KCC2, GAD67, and MeCP2 E1 expression in the adult cortex of female Rett and control mice. (**A**) Expression in p33-pre-symptomatic mice (control vs. pre-symptomatic Rett mice). Vinculin was used as a loading control. (**A**) Expression in six-months old mice (control vs. symptomatic Rett mice). Both Western blots (**A**,**B**) are cropped stripes of two different membranes each, and incubated with each antibody separately (target protein and loading control respectively). Each lane is a different animal. Representative blots shown.

**Figure 4 ijms-21-00518-f004:**
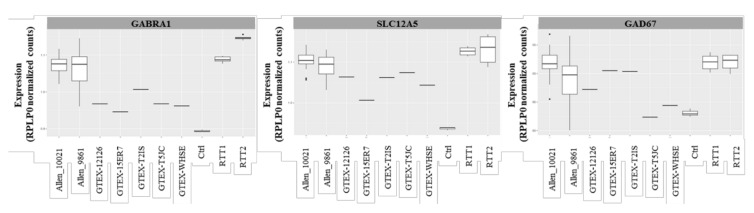
GABAergic pathways genes expression assessment in brain patients by RNAseq. Comparison of the GABA ionotropic receptors subunits *GABRA1*, and KCC2 and GAD67 genes between the patients (RTT 1 and RTT 2), the intra-assay control (Ctrl), and publicly available controls, showing a tendency to increase in the Rett patients. The results were normalized by *RPLP0*.

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
