# Peer review of "Comprehensive Analysis of GABAA-A1R Developmental Alterations in Rett Syndrome: Setting the Focus for Therapeutic Targets in the Time Frame of the Disease"

_ijms, 2020, doi:10.3390/ijms21020518_

Round 1

Reviewer 1 Report

In this work, Oyarzabal and coauthors investigate the expression of GABA receptor subunits in different models of Rett syndrome: they use a mouse neuroblastoma cell line to show a link between MECP2 presence and Gabra1 expression upregulation; they use Mecp2-/+ mouse cortex and observe a differential expression of both GABAA1R and KCC2 depending on the age of the animal and finally they use post-mortem brain samples (cortex) to reveal an increase in the expression of GABRA1, KCC2 and GAD67 when compared to healthy controls.

Stimulatory/inhibitory imbalance is an emerging hallmark of Rett syndrome but it is still unsufficiently understood. This work provides some insights into the relationship between MECP2 function and GABA pathway regulation, pointing to the importance of developmental timing when studying the impact of MECP2 loss of function in the disease. Although of potential interest to the readership of IJMS, I have some concerns related to the experimental set up that must be explained before this manuscript is suited for publication.

Major points:

-          The Mecp2 knockdown experiment in mouse cells and restitution with the human gene in Figure 1 is confusing: while the shRNA used seems to work reasonably well (figure 1A, left), quantitation of the human MECP2 shown in Figure 1A right is unexpected: what is exactly being measured in the control condition (where the human MECP2 present should be zero), why is the h-MECP2 decreasing with the shRNA (that is supposed to be specific for the endogenous mouse mRNA sequence), why is the overexpression only 2-fold and why is the fourth condition (shRNA+hMECP2mut) the one with the lowest levels of h-MECP2 (when three of the conditions are expected to be roughly the same)? In the same samples, the authors analyse m-Gabra1 and see a spectacular increase of expression of several orders of magnitude (Figure 1B, left). One might think that the panels are actually swaped and Figure 1A right corresponds to Figure 1B left, and viceversa. However, the fact that m-Gabra1 is virtually absent in N2A cells also makes this quite implausible. All this issues need to be clarified so that the reader understands the experimental setting.

-          Since the mRNA levels do not always reflect changes at the protein level, Western Blot against the MECP2 protein should rather be shown for the different conditions in Figure 1. IF analysis in Figure 2 only displays one transfected cell per image and is not sufficient to evaluate the efficiency of the knockdown.

-          The experimental design for the RNAseq data is confusing: only 2 RTT patients were used, and frontal and occipital cortex regions were apparently pooled in the analysis (line 411). These samples were compared to several publicly available controls in which male ¿! were included, and for which not the same two regions were analyzed, some being pools of frontal+temporal and some being just temporal cortex (line 419). In addition to this somewhat convoluted experimental system, the results are very difficult to read in Figure 4. Resolution of the image needs to be improved and justification for the choice of the different controls is needed.

Minor points:

-          Line 429, the sentence is not finished: “…RTT patients were averaged due to the high.”

-          Supplementary Figure 1 should be referenced to in the main text.

Reviewer 2 Report

Oyarzabal and colleagues aim at studying the relationship between MeCP2 and GABA-A1 receptor expression. This study is motivated by the well-established alteration in excitatory/inhibitory balance in Rett syndrome patients and animal models. Rett syndrome is caused by mutations in the mecp2 gene.

The authors attempt to measure the expression of GABA receptor subunit A1 in cell lines and primary neuronal cultures upon manipulation of MeCP2 levels and in tissue from MeCP2 mouse knock-outs and patient samples. However, I have several concerns regarding the experimental design and data presentation:

Figure 1: what is the rationale for choosing this particular mutant?

Figure 1a: when using the qPCR primers against human MeCP2 primers (right side graph), it is surprising that amplification occurs in N2A and shRNA conditions. And why is the bar in the shRNA+hMeCP2mut condition so low? These issues put in question the specificity of the primers. The confirmation of the exogenous expression of hMeCP2 should have been done additionally by western blot analysis.

Figure 1b: the expression levels of gabra1 are calculated relative to which condition? qPCR is often used as a relative quantification method. It is not clear which conditions are being compared here.

Figure 2: the effect of MeCP2 knockdown in GABRA1 levels can not be concluded from just one picture. The authors should quantify the signal intensity from several cells and independent cell preparations and perform statistical analysis. Or preferably, perform western blot analysis.

Figure 4: It is not possible to read the diagram.

Round 2

Reviewer 1 Report

After reading the revised manuscript, I appreciate the clarifications the authors have added, specially regarding Figure 1 and the analysis of the RNAseq data. Even though some of the quantitations are still somewhat unclear, I am satisfied with the justification that background amplification of the mouse form is interfering with the human gene, as stated now in the text. 

Author Response

Thank you very much for, once again, your consideration on reading the manuscript and for your comments on it.

We have added a couple of sentences in the text (in blue, lines 473 and 194) regarding the quantitation in ICCs, hopping we have made it clearer. 

Again, thank you very much for your kind review of the manuscript, 

Best regards. 

Reviewer 2 Report

Some of my comments remain to be address. New experiments need to be performed:

Due to the lack of specificity of the qPCR primers the expression of exogenous hMeCP2 remains to be validated. As previously suggested, this should be done using western blot. I acknowledge that the transfection efficiency of neuronal cultures is usually very low. However, the staining needs to be repeated in independent culture preparations.

Author Response

Thank you very much for, again, reading the manuscript and for your comments on it.

We understand your concerns regarding the qPCR and primers specificity, as they have been, indeed, pointed out by another reviewer. While we see how a western blot will back up the results, we are unfortunately not in the disposition of performing it on the shortest term.

We hope (based on the previous reviewer satisfaction with the result) that the explanation and the changes on the text have cleared things up. 

Anyhow, we undoubtedly thank you for your comments and concerns on that figure, as we share the same point of view. 

Regarding the quantitation of the ICCs, the measurements have been done in a n=10 neurons from two different primary cultures. As with that we obtained a robust reduction we decided not to perform more experiments, thus minimizing the number of animals sacrificed for such term. We have realized based on your comment, which we appreciate, that such number and fact hadn't been stated throughout the text. We have done it now in the figure legend as well as in the methods section (written in blue, lines 194 and 473).

Hopping we have cleared things up with our comments,

And again, thanking you very much your comments,

Best regards,

A. 

Best regards.

Round 3

Reviewer 2 Report

My comment concerning the quantification of the ICC has been addressed. I am still concerned that the expression of exogenous MeCP2 is not validated at the protein level, however I think that cDNA validation may be sufficient to allow publication.